

# R-DECO: an open-source Matlab based graphical user interface for the detection and correction of R-peaks

Jonathan Moeyersons[1], Matthew Amoni[2,3], Sabine Van Huffel[1], Rik Willems[2,3] and Carolina Varon[1]

[1] STADIUS Center for Dynamical Systems, Signal Processing and Data Analytics, Department of Electrical Engineering (ESAT), KU Leuven, Leuven, Belgium
[2] Department of Cardiovascular Sciences, KU Leuven, Leuven, Belgium
[3] Department of Cardiology, University Hospitals Leuven, Leuven, Belgium

## ABSTRACT

Many of the existing electrocardiogram (ECG) toolboxes focus on the derivation of heart rate variability features from RR-intervals. By doing so, they assume correct detection of the QRS-complexes. However, it is highly likely that not all detections are correct. Therefore, it is recommended to visualize the actual R-peak positions in the ECG signal and allow manual adaptations. In this paper we present R-DECO, an easy-to-use graphical user interface (GUI) for the detection and correction of R-peaks. Within R-DECO, the R-peaks are detected by using a detection algorithm which uses an envelope-based procedure. This procedure flattens the ECG and enhances the QRS-complexes. The algorithm obtained an overall sensitivity of 99.60% and positive predictive value of 99.69% on the MIT/BIH arrhythmia database. Additionally, R-DECO includes support for several input data formats for ECG signals, three basic filters, the possibility to load other R-peak locations and intuitive methods to correct ectopic, wrong, or missed heartbeats. All functionalities can be accessed via the GUI and the analysis results can be exported as Matlab or Excel files. The software is publicly available. Through its easy-to-use GUI, R-DECO allows both clinicians and researchers to use all functionalities, without previous knowledge.

## INTRODUCTION

The electrocardiogram (ECG) is one of the primary screening and diagnostic tools of the cardiologist. It records the electrical activity of the heart, which generates the myocardial contractions. A crucial step in the study of the ECG is the location of the QRS-complexes. As can be seen in Fig. 1, these complexes are the most prominent waveforms in the ECG. They contain an enormous amount of information about the state of the heart. This is why the detection of the QRS-complexes constitutes the basis for almost all automated ECG analysis algorithms (*Kohler, Hennig & Orglmeister, 2002*). Once these have been identified, more elaborated analyses can be performed, such as heart rate variability (HRV).

Corresponding author
Jonathan Moeyersons,
Jonathan.Moeyersons@esat.
kuleuven.be

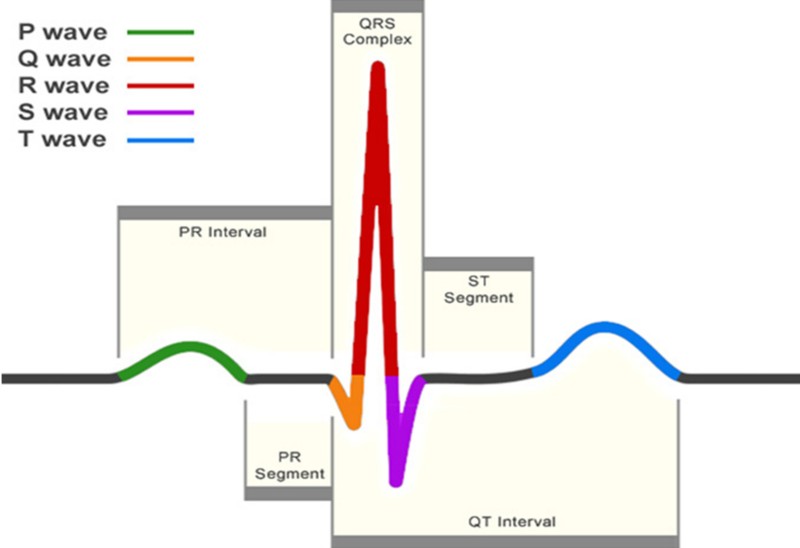

**Figure 1** **A normal heartbeat as recorded by an ECG.** The QRS-complex can be observed in the center. The detection of this complex is crucial for almost all ECG analysis algorithms.

Four decades of automated QRS detection research has resulted in a variety of methods using different approaches. These methods can be stratified based on derivatives, digital filters, wavelet-transforms, classifiers, etc. (*Pan & Tompkins, 1985*; *Dohare, Kumar & Kumar, 2014*; *Fujii et al., 2013*; *Sharma & Sunkaria, 2016*; *Chen, Chen & Chan, 2006*). Despite the wide methodological variety, most of these QRS detectors have the same algorithmic structure. This can be divided in two steps: pre-processing and decision making (*Kohler, Hennig & Orglmeister, 2002*).

In the pre-processing step the QRS-complex is highlighted and the other signal components are suppressed to facilitate the detection. The resulting signal is then used to detect the occurrence of QRS-complexes in the decision making step. This is done by using either fixed or adaptive thresholds. Despite high detection rates, some QRS-complexes remain undetected. Reasons for this might be small amplitudes, wide complexes or contamination by noise (*Arzeno, Deng & Poon, 2008*). Therefore, in many algorithms an extra post-processing step is added for the exact determination of the temporal location of the detected QRS-complex.

One of the most established QRS detection algorithms is the Pan–Tompkins algorithm (*Pan & Tompkins, 1985*). Although it was developed in the eighties, it achieves comparable performance to many more elaborate algorithms (*Elgendi et al., 2014*). In this paper, an envelope-based procedure that enhances the QRS-complexes and flattens the rest of the ECG is used in combination with an adapted version of the threshold-based approach of the Pan–Tompkins algorithm. This method, which was proposed by our group in *Varon et al. (2015)*, combines the simplicity of an envelope-based procedure, while maintaining the accuracy of many more elaborate methods.

In a review paper, *Elgendi et al. (2014)* have compared the results of 22 beat detection algorithms on the MIT-BIH arrhythmia database. When comparing the results of the automated algorithms with expert annotations, they have shown that many algorithms obtained excellent accuracy. However, none of the algorithms reached perfection. This means that, no matter how good the QRS detection algorithm is, it is highly likely that not all annotations are correct. Therefore, it is recommended to visually inspect and review each signal before further analysis (*Pichot et al., 2016*).

Many of the existing ECG toolboxes have focussed on the derivation of HRV-analysis parameters from RR-intervals, the time between subsequent R-peaks. This makes sense, since most of the available hardware include some kind of QRS-complex detection algorithm. However, this does not necessarily mean that the output of these devices are the raw RR-intervals. Many of these devices have a built-in post-processing algorithm, which compensates for false detections by averaging over a certain range of RR-intervals (*Niskanen et al., 2004*; *Pichot et al., 2016*; *Vicente et al., 2013*). However, for some analyses, such as ECG derived respiration (EDR) or beat-to-beat variability of repolarization (BVR), it is of utmost importance that the actual R-peak of the QRS-complex is detected. Therefore, it is necessary to visualize the actual R-peak positions in the ECG signal and allow the possibility to make manual adaptations.

In this paper, we present R-DECO, a MATLAB-based, graphical user interface (GUI) for the detection and correction of R-peaks. This user interface includes the developed R-peak detection algorithm and provides the user with the possibility to correct possible false detections in a very straightforward way. R-DECO was developed by the biomedical data processing research team (BIOMED) at the Department of Electrical Engineering (ESAT) of KU Leuven. The software is freely available for Windows operating systems at https://gitlab.esat.kuleuven.be/biomed-public/r-deco.

The objective of this paper is to provide a detailed description of R-DECO, including the proposed R-peak detection algorithm and an overview of the different possibilities of this new software.

## COMPUTATIONAL METHODS

### R-peak detection

We developed an R-peak detection algorithm that is based on an enveloping procedure. It achieves a 99.60% sensitivity and 99.69% positive predictive value (PPV) on the MIT/BIH Arrhythmia Database (*Moody & Mark, 2001*). The algorithm can be divided in three steps: pre-processing, decision and post-processing.

#### Pre-processing

The pre-processing consists of an enveloping procedure, which enhances the QRS-complexes and flattens the rest of the ECG (*Varon et al., 2015*). A visual explanation of the method is shown in Fig. 2.

First, the upper ($U$) and lower ($L$) envelopes are computed from the ECG signal by the secant method. This method selects the segment with the steepest positive and negative slope in a user-defined window with length $t$. Once $U$ and $L$ are obtained, they are used to

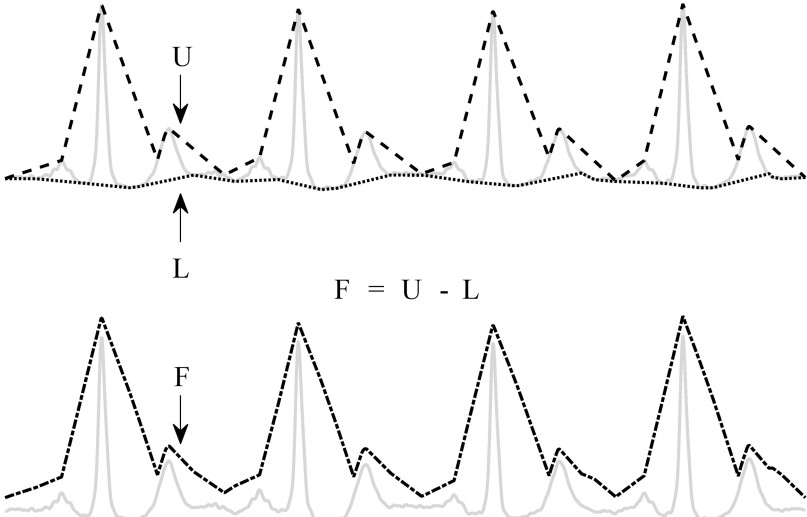

**Figure 2 Enveloping procedure.** The flattened ECG (F) is constructed by subtracting the lower envelope (L) from the upper envelope (U). This enhances the QRS-complex and flattens the rest of the ECG signal.

derive a flattened version of the ECG signal ($F$): $F = U - L$. Since $L$ is subtracted from $U$, the baseline is eliminated and only a positive signal, $F$, remains.

### Decision

The locations of the QRS-complexes are found by detecting the peaks in the flattened ECG. These peaks are detected in three stages. First, all samples with an amplitude lower than the amplitude of the sample 80 ms further are selected. The 80 ms step size was experimentally defined. This results in the selection of the upward slopes. As a second step, only the upward slopes that are longer than the step size are selected in order to exclude small peaks. Finally, the maximum is selected in a window, with a length equal to the step size, that starts from the last selected sample of the upward slope. A graphical representation of this process is shown in Fig. 3.

On this selection of peaks, the adaptive thresholding procedure of the Pan–Tompkins algorithm is applied to define the peaks that correspond to the QRS-complexes.

### Post-processing

The thresholding procedure generally produces satisfactory results for the detection of the QRS-complexes. However, some of the automatically generated RR-intervals might be physiologically unreasonable and need to be removed for further analysis. A slightly modified version of the search-back procedure as proposed in *De Chazal et al. (2003)* was used for this purpose.

Once the positions of the QRS-complexes are identified, the original ECG is used to find the exact location of the R-peaks. The search for an R-peak is performed up to 50 ms from the peak in the flattened signal. This extra search is necessary because the presence of large S-waves might shift the peak in the flattened signal toward the valley of the S-wave.

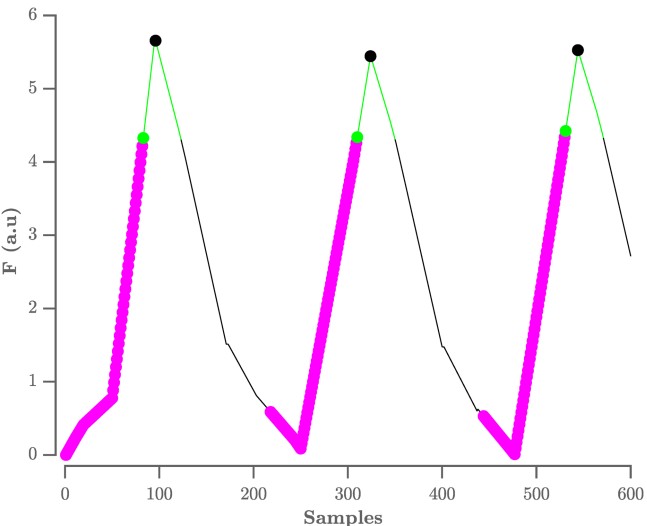

**Figure 3 Procedure to select R-peaks.** The resulting flat ECG denoted F is indicated by the black line. The samples with an amplitude lower than the sample 80 ms further are indicated by the magenta circles, with the last sample indicated by the green circle. The search window is indicated by the green line. The selected R-peaks are indicated by the black circles. A.u. stands for arbitrary units.

### Evaluation on the physionet MIT/BIH arrhythmia database

We used the MIT/BIH arrhythmia database to evaluate the proposed algorithm (*Moody & Mark, 2001*). This dataset consists of 48 half-hour ECG signals, which were recorded in the Boston's Beth Israel Hospital between 1975 and 1979. All recordings were annotated by two independent cardiologists who also made a distinction between normal and abnormal beats. In total, 110,122 heartbeats were annotated, of which 89,133 were labeled as normal. Each recording contains two channel ECG signals with a sampling frequency of 360 Hz. In most records, one channel is lead II and the other channel is V1. However, we only used the first channel for the evaluation.

As mentioned previously, the pre-processing consists of a flattening step of the ECG with a user-defined window width. To evaluate the sensitivity of the performance to the choice of the width we have tested multiple window widths. As can be observed from Fig. 4, comparable results were obtained for window widths between 250 and 350 ms.

In Table 1 we list the R-peak detection results of the proposed algorithm with an envelope width of 300 ms and without post-processing. We obtained an overall sensitivity of 99.60% and PPV of 99.69%. When including the post-processing, we obtained an overall sensitivity of 99.09% and PPV of 99.80%. These results are comparable with those in literature, especially with the Pan–Tompkins algorithm, which reaches a sensitivity of 99.76% and a PPV of 99.56% (*Elgendi et al., 2014*).

While these results are very promising, we can also observe that for some recordings only moderate detection results are achieved. This decrease in performance is generally due to loss of signal, unusual morphology or stretches of extremely irregular rhythms. For instance, recording 116 and 208 contain stretches where the signal is lost in the first channel. However, the recordings with the highest amount of false detections are 108, 203

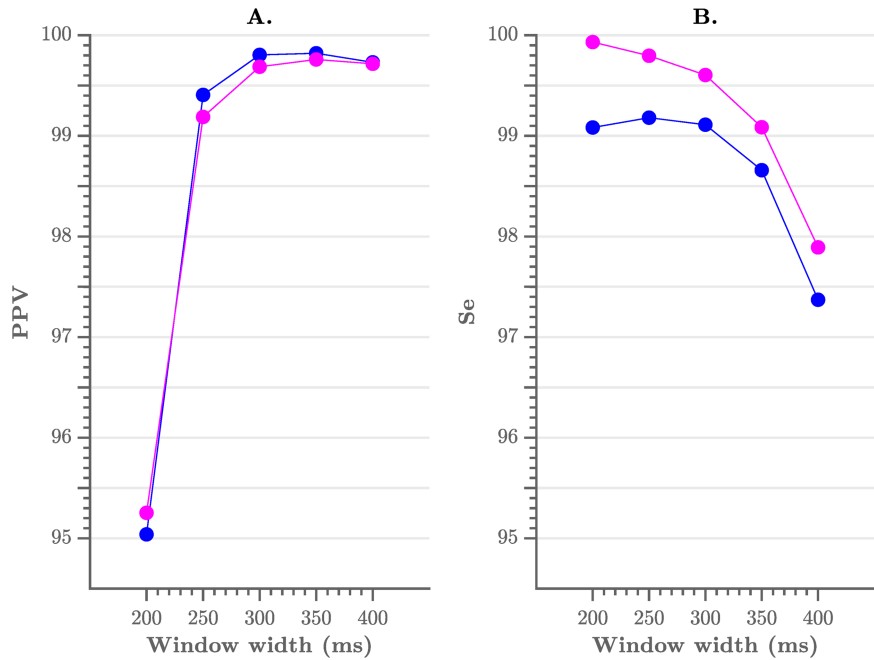

**Figure 4 Performance of the algorithm, compared to the choice of window width.** (A) The positive predictive value and (B) the sensitivity of the algorithm. A window width between 250 and 350 ms results in the best performance.

and 210. Correctly detecting the R-peaks in recording 108 has been proven difficult for many algorithms (*Pan & Tompkins, 1985*). It contains a lot of baseline wander and additionally, very tall and sharp P-waves. These characteristics make it difficult to distinguish P-waves from R-peaks and thus result in a high false positive count. However, the highest amount of false positives is observed in recording 203 (92.25% sensitivity). This might be explained by the extremely high percentage of premature ventricular contractions (PVC) present in the recording, almost 15%. Since the envelope width was fixed during the detection process, one may assume that the performance could be improved if manual adjustments were permitted. This holds as well for other records with PVC's, such as record 210.

The noise tolerance of the algorithm was evaluated with the MIT-BIH Noise Stress Test Database (*Moody, Muldrow & Mark, 1984*). We observed that both median PPV and sensitivity remained around 100% above a signal-to-noise-ratio of six dB. From this threshold the performance of the algorithm decreased significantly.

From the analysis of the results, we could deduce two main factors that influence the results of the algorithm: (1) the envelope width and (2) the RR-post processing. The number of samples in the envelope is important, since it can be regarded as a filter of the RR-intervals. Smaller envelope widths might result in the enhancement of more peaks than only the R-peaks. This might be beneficial in the case of small R-peaks, but might also enhance artefact peaks. Larger envelope widths might cause adjacent R-peaks to be merged in the flattened signal. In practice, this might result in the failure of detecting premature heartbeats, which appear shortly after the previous heartbeat. In summary,

**Table 1  Performance of the R-peak detection algorithm on the Physionet MIT/BIH dataset.**

| Record | Total (beats) | TP (beats) | FP (beats) | FN (beats) | Se (%) | PPV (%) |
|---|---|---|---|---|---|---|
| 100 | 2,273 | 2,273 | 0 | 0 | 100 | 100 |
| 101 | 1,865 | 1,864 | 4 | 1 | 99.95 | 99.79 |
| 102 | 2,187 | 2,187 | 0 | 0 | 100 | 100 |
| 103 | 2,084 | 2,084 | 0 | 0 | 100 | 100 |
| 104 | 2,229 | 2,227 | 3 | 2 | 99.91 | 99.87 |
| 105 | 2,572 | 2,542 | 41 | 30 | 98.83 | 98.41 |
| 106 | 2,027 | 2,023 | 2 | 4 | 99.80 | 99.90 |
| 107 | 2,137 | 2,130 | 0 | 7 | 99.67 | 100 |
| 108 | 1,763 | 1,739 | 80 | 24 | 98.64 | 95.60 |
| 109 | 2,532 | 2,532 | 0 | 0 | 100 | 100 |
| 111 | 2,124 | 2,123 | 2 | 1 | 99.95 | 99.91 |
| 112 | 2,539 | 2,539 | 0 | 0 | 100 | 100 |
| 113 | 1,795 | 1,795 | 0 | 0 | 100 | 100 |
| 114 | 1,879 | 1,877 | 6 | 2 | 99.89 | 99.68 |
| 115 | 1,953 | 1,953 | 0 | 0 | 100 | 100 |
| 116 | 2,412 | 2,388 | 2 | 24 | 99 | 99.92 |
| 117 | 1,535 | 1,535 | 0 | 0 | 100 | 100 |
| 118 | 2,278 | 2,278 | 1 | 0 | 100 | 99.96 |
| 119 | 1,987 | 1,987 | 2 | 0 | 100 | 99.90 |
| 121 | 1,863 | 1,861 | 12 | 2 | 99.89 | 99.36 |
| 122 | 2,476 | 2,476 | 0 | 0 | 100 | 100 |
| 123 | 1,518 | 1,518 | 0 | 0 | 100 | 100 |
| 124 | 1,619 | 1,619 | 0 | 0 | 100 | 100 |
| 200 | 2,601 | 2,595 | 8 | 6 | 99.77 | 99.69 |
| 201 | 1,963 | 1,958 | 0 | 5 | 99.75 | 100 |
| 202 | 2,136 | 2,120 | 14 | 16 | 99.25 | 99.34 |
| 203 | 2,980 | 2,749 | 20 | 231 | 92.25 | 99.28 |
| 205 | 2,656 | 2,641 | 2 | 15 | 99.44 | 99.92 |
| 207 | 1,860 | 1,855 | 8 | 5 | 99.73 | 99.58 |
| 208 | 2,955 | 2,941 | 2 | 14 | 99.53 | 99.93 |
| 209 | 3,005 | 3,005 | 0 | 0 | 100 | 100 |
| 210 | 2,650 | 2,582 | 3 | 68 | 97.43 | 99.88 |
| 212 | 2,748 | 2,748 | 1 | 0 | 100 | 99.96 |
| 213 | 3,251 | 3,250 | 0 | 1 | 99.97 | 100 |
| 214 | 2,262 | 2,259 | 3 | 3 | 99.87 | 99.87 |
| 215 | 3,363 | 3,354 | 0 | 9 | 99.73 | 100 |
| 217 | 2,208 | 2,202 | 0 | 6 | 99.73 | 100 |
| 219 | 2,154 | 2,154 | 1 | 0 | 100 | 99.95 |
| 220 | 2,048 | 2,047 | 0 | 1 | 99.95 | 100 |
| 221 | 2,427 | 2,425 | 2 | 2 | 99.92 | 99.92 |
| 222 | 2,483 | 2,475 | 21 | 8 | 99.68 | 99.16 |
| 223 | 2,605 | 2,605 | 0 | 0 | 100 | 100 |

| Record | Total (beats) | TP (beats) | FP (beats) | FN (beats) | Se (%) | PPV (%) |
|---|---|---|---|---|---|---|
| 228 | 2,053 | 2,044 | 59 | 9 | 99.56 | 97.19 |
| 230 | 2,256 | 2,256 | 0 | 0 | 100 | 100 |
| 231 | 1,571 | 1,571 | 0 | 0 | 100 | 100 |
| 232 | 1,780 | 1,780 | 17 | 0 | 100 | 99.05 |
| 233 | 3,079 | 3,070 | 0 | 9 | 99.71 | 100 |
| 234 | 2,753 | 2,753 | 1 | 0 | 100 | 99.96 |
| Total | 109,494 | 108,989 | 317 | 505 | 99.60 | 99.69 |

a larger envelope width results in less false positives and more false negatives and the opposite is true for a small envelope width. A similar effect can be observed when the RR-intervals are post-processed. This increases the certainty of detection of the algorithm and thus results in less false positives. The downside is that, in the presence of abnormal rhythms, it also results in more false negatives.

# SOFTWARE DESCRIPTION

The algorithms have been implemented with MATLAB R2018a. We used GUIDE, MATLAB's GUI development environment, to design the GUI of R-DECO. The current subsection describes the possible input data formats and the user interface.

## Input data formats

The standard input of the toolbox is raw or filtered ECG data. This can be both single- or multichannel ECG. Since a plethora of open formats exist for storing the ECG, it would be impossible to write supporting software for all formats (*Niskanen et al., 2004*). Therefore, we focussed on the data formats that are most commonly used by our clinical partners in the cardiology department of the UZ Leuven, Belgium. The following file formats are supported:

- ISHNE-Holter files (*.ecg)
- MATLAB files (*.mat)
- European Data Format (*.edf)
- Text files (*.txt)
- Excel files (*.xls or *.csv)

An ISHNE-Holter file is organized in a header record, followed by a data block that contains all digital ECG samples. This file format was developed to facilitate data exchange and research in the field of Holter (*Badilini, 1998*). The software automatically extracts all ECG channels and also the sampling frequency.

A MATLAB formatted file can contain one variable, up to an entire workspace. Therefore, if the file contains more than one variable, the user is prompted to select the variable containing the ECG signal. In the specific case that the selected file is a structure, the software allows to search within the structure until the ECG signal is selected.

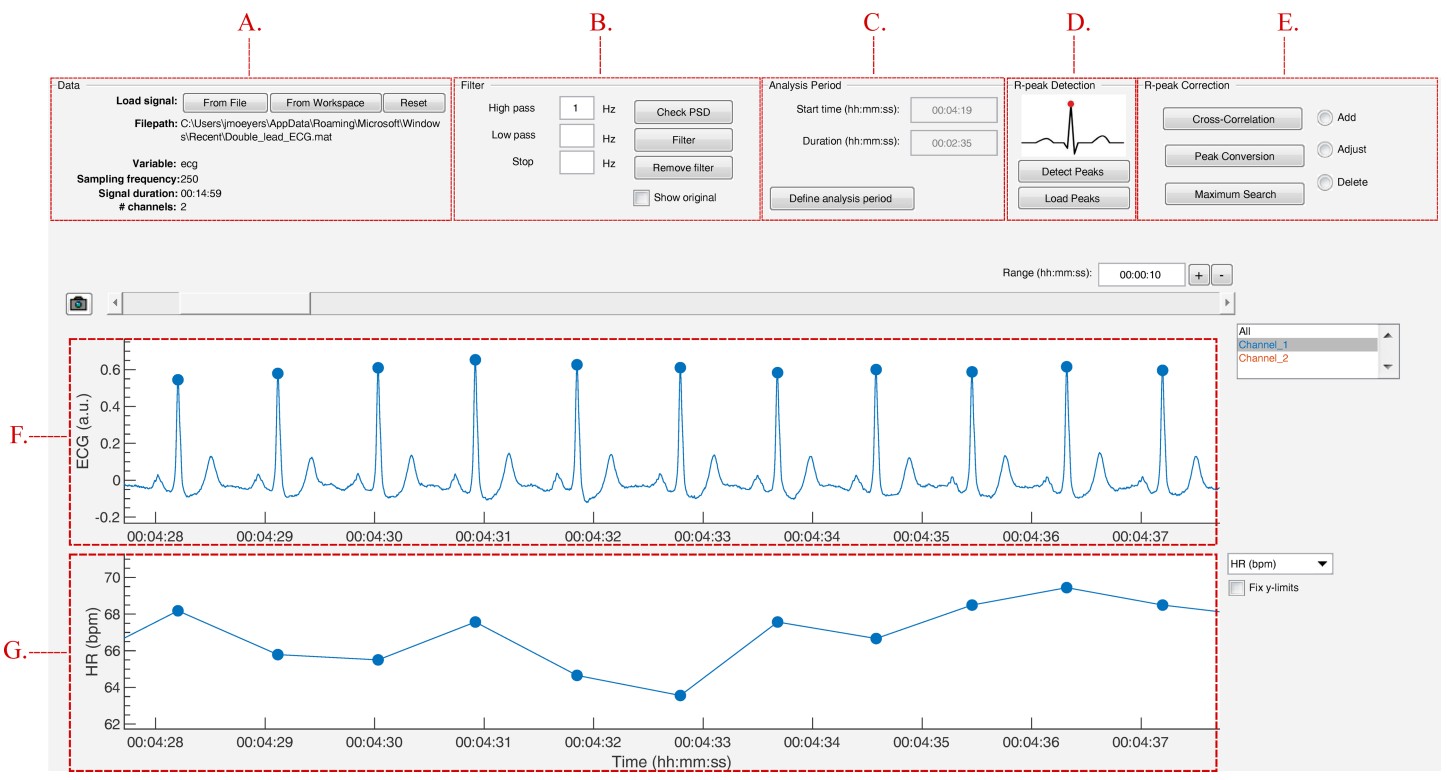

**Figure 5 The graphical user interface of R-DECO.** The user interface can be divided in five segments: (A) Data, (B) Filter, (C) Analysis Period, (D) R-peak Detection and (E) R-peak Correction. The ECG signal and the resulting tachogram are shown in respectively (F) and (G). The detected R-peaks are depicted as small blue circles.

A standard European Data Format, EDF, file consists of a header record and the data records (*Kemp et al., 1992*). It was originally intended for the digital storage and exchange of EEG and polysomnogram recordings, but currently it can store a variety of annotations and signals, such as EMG, ECG and many more (*Kemp & Olivan, 2003*). Since not all EDF files have the same standard labels, the user is prompted to identify the ECG channel(s). Additionally, the software attempts to identify the sampling frequency of the selected signal by scanning the file.

As an extra feature, the software allows the user to load a session. When the current session is interrupted, the session can be saved as a MATLAB file. It includes all the analysis parameters, the ECG signal and the RR-intervals, if computed. When a previous session is loaded, the software restores the entire user interface to the moment on which the session was saved. This allows the user to pause and continue, whenever wanted.

### User interface

The strength of this toolbox is that everything is operated through a single GUI. As shown in Fig. 5, it can be divided in five segments: Data, Filter, Analysis Period, R-peak Detection and R-peak Correction. All segments are described below.

### Data

In the data panel, the user has the option to load data in two different ways: from a file or from the MATLAB workspace. Both can be accessed via the respective pushbuttons.

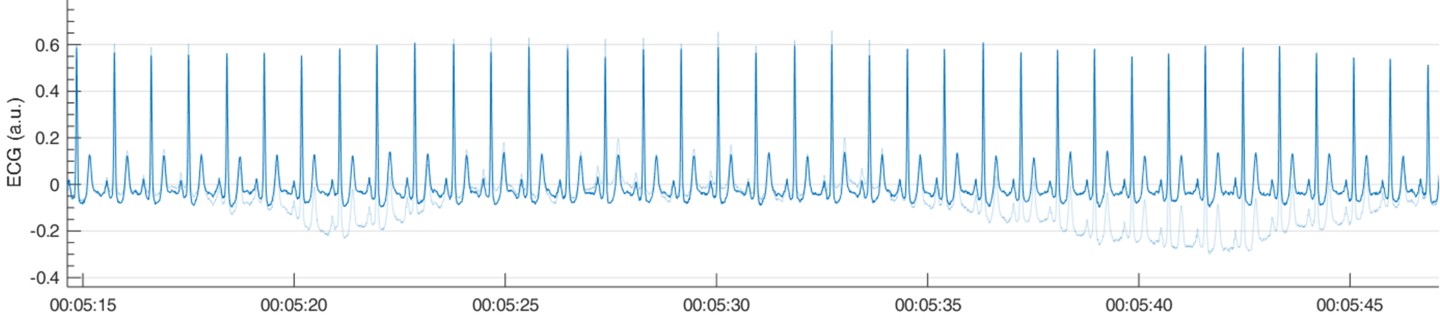

**Figure 6 Example of a high pass filter.** By clicking the "Show Original" checkbox, the original signal (light blue) is overlaid on the filtered signal (blue).

When a file is selected, the software visualizes a small segment of the ECG signal. If the signal is inverted, the user can indicate this and the file will be inverted before further analysis. The software also scans the selected file for the sampling frequency. If this is not found, the software prompts the user to manually indicate the sampling frequency.

Finally, the data panel also contains a reset button. This button allows the user to reset the entire GUI. It empties all plots, restores all default variables and deletes all results. If the user has not yet saved the current analysis, the user is prompted to confirm the reset action to prevent unwanted loss of information.

### Filter

Since ECG signals can be contaminated with noise, filtering is often essential for further analysis. R-DECO provides three basic filters: high pass, low pass and a notch filter.

The high pass filter consists of a zero phase, second order Butterworth filter. The low pass filter consists of a zero phase, fourth order Butterworth filter. Finally, a zero phase notch filter is also included. The latter could be used to remove the power-line interference. Important to note is that filtering actions are always executed on the original signal to ensure repeatability.

In order to aid the user in the selection of appropriate frequency threshold(s), R-DECO is able to compute and display the power spectrum. An estimate of the power spectrum is computed using the Welch method (*Welch, 1967*). As a default, we used a window of 500 samples with 60% overlap.

To visualize the effect of the filtering in the frequency domain, R-DECO displays both the filtered and the original power spectrum. Furthermore, the effect of the filtering in the time domain can also be investigated by checking the "Show Original" checkbox. This will overlay the original signal on top of the filtered signal (Fig. 6).

### Analysis period

In the Analysis Period panel the user has the possibility to define an analysis window. After pushing the "Define analysis period" button, the user has to select a window by clicking, dragging and releasing the mouse. The window is shown as a transparent patch over the data and can be enlarged, shrinked or moved with the mouse. After the initial window

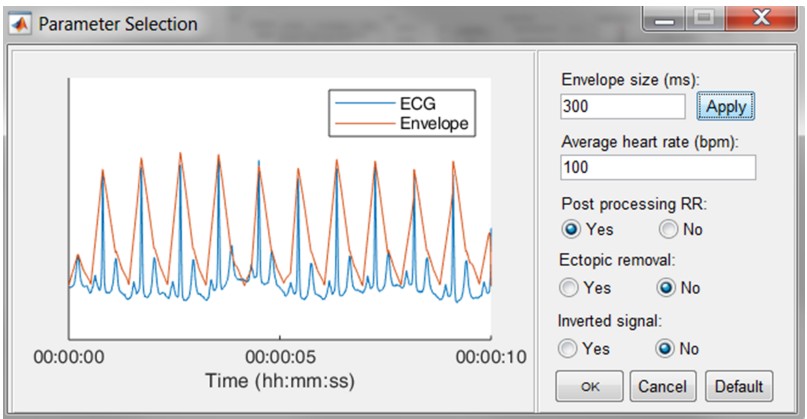

**Figure 7 The R-peak parameter selection window.** The user can adjust the envelope size, the average heart rate and indicate if RR post-processing is necessary. Pressing the Default button restores the default values.

is drawn, the user can further modify the analysis window by changing the start time and/or the duration of the window.

When a window is selected, the user can accept the analysis window by pressing the "Apply changes" button. This prompts the *x*-limits of the graph to match the analysis window and disables the window modifications. From here on, all further analysis will be performed solely on the selected window.

All the above is very useful when the to-be-selected time period is known in advance, but this is not always the case. Sometimes the selection of the analysis window is depending on certain patterns in the tachogram, hence the tachogram has to be constructed first. Therefore, if R-peaks have been detected already, the user is prompted to indicate whether he/she would like to keep the detected R-peaks.

### R-peak detection

The execution of the R-peak detection algorithm, as described in the first section, is initiated when the "Detect Peaks" button is pushed. However, before the actual algorithm is executed, the user is able to adjust the default parameters of the algorithm.

In Fig. 7, an epoch of 10 s of the ECG signal and its respective enveloped signal is shown as an example of the flattening procedure. The user can adjust the envelope size to the desired value and enact the changes by pressing the "Apply" button. Additionally, an estimation of the average heart rate can be defined to compute the boundaries of normal-to-normal RR-intervals. Lastly, the user can select the automatic post-processing of the RR-intervals.

Since some devices have built-in QRS detection algorithms and some researchers have their own preferred QRS detection algorithm, the software allows to load R-peak locations. These will be displayed the same way the R-peaks of the algorithm are displayed.

### R-peak correction

In case of HRV studies, only the normal-to-normal RR-intervals need to be taken into account. This can be achieved by selecting the ectopic removal option. This option corrects ectopic beats, without altering the normal RR-intervals.

After finishing the detection process, either by detecting or loading the R-peaks, it is still possible that not all R-peaks are accurately detected. In R-DECO, the user can make manual and (semi-)automatic adjustments to the R-peak locations.

The manual methods are: add, adjust and delete. These methods can be activated by selecting the specific radiobuttons or via a context menu, which is linked to each R-peak. These manual methods allow the user to correct wrong or missing annotations either in all or in individual leads.

- Add: When this radiobutton is active, new R-peaks can be added by clicking in the ECG graph. The program selects the signal sample that is closest to the mouse position in a symmetric window of 300 ms around the mouse position. Upon mouse release, a new R-peak is added and the tachogram is adapted.
- Adjust: While hovering over the ECG graph, the R-peak closest to the mouse location is selected. After clicking on the desired R-peak, it can be moved by dragging the mouse. The movement of the selected R-peak is restricted by the previous and next R-peak. While adjusting the R-peak location, the tachogram is automatically updated. Upon mouse release, the new R-peak location is saved.
- Delete: While hovering over the ECG graph, the R-peak closest to the mouse location is selected. After clicking on the desired R-peak, more to-be-deleted R-peaks can be selected by dragging the mouse. Upon mouse release, the selected R-peaks are removed and the tachogram is adapted.

The three (semi-)automatic R-peak correction methods are: cross-correlation, peak conversion and maximum search.

- Cross correlation: For this method, a symmetrical window of 300 ms around each R-peak is selected. Then, all heartbeats are normalized by subtracting the mean and dividing it by the standard deviation. Then, a trimmed average QRS-complex is computed of all the positive and "negative" R-peaks. In this work a "negative" R-peak is understood as the absence of an R-peak or the presence of a very prominent S-wave, also described as RS-complex.
- The user is prompted to select either the positive or the "negative" average heartbeat. If necessary, the user can also adjust the location of the R-peak on the selected template. This is all graphically displayed as shown in Fig. 8. Finally, the cross-correlation of every heartbeat is computed with the trimmed average and the R-peak is re-located, based on the highest correlation value.
- Peak conversion: The absolute amplitude of every R-peak annotation is compared with the previous and next sample's absolute amplitude. If it is bigger than the previous and smaller than the next, the location of the R-peak will be shifted forward, until an extremum is obtained. If it is the other way around, the location of the R-peak will be shifted backwards.
- This functionality avoids the "jumping" of R-peaks from, for example, an actual R-peak to a pre-mature ventricular contraction, which might be the case when window search is

applied. Furthermore, pressing the button more than once will not affect the relocation after one correction.

- Maximum search: Firstly, a symmetric window of 300 ms around each R-peak is selected. Secondly, the user is prompted to select either the maximum, minimum or absolute maximum. Based on this selection, the respective extremum within the window is selected as new R-peak location.

## Save and export results

By default, the results are saved as a MATLAB file. This file includes the R-peak locations and the RR-intervals, which can further be used for HRV-, BVR- or any other analysis that requires R-peak detection. Additionally, the software can export the results in two different ways: (1) an Excel file and (2) a MATLAB file.

(1) Excel file (*.xls): A new workbook is created with a general overview of the file on the first sheet: the number of channels, the sampling frequency, the duration of the signal and the duration of the analysis period. The number of additional sheets is defined by the number of channels, since for every channel, a separate sheet is created. This contains the R-peak locations, the RR-intervals and a number of basic metrics, such as the mean heart rate.

(2) MATLAB file (*.mat): This file contains a single structure named *data*. In accordance to the structure of the Excel file, a structure is created per channel. This contains the signal in the analysis window, the R-peak locations and the RR-intervals. This option is especially useful for further analysis in MATLAB.

## Data browser

In order to graphically correct the R-peaks it is important to have a clear view of the segment to be investigated. Therefore, after the R-peaks are detected, the software immediately displays the ECG signal with the R-peak annotations and the respective tachogram, as can be seen in Fig. 9.

The window width of the *x*-axis can be adjusted in three different ways: (1) the range edit box, (2) the plus and minus buttons or (3) by using the zoom button. All three methods also adjust the width of the scroll bar.

The scroll bar can be used to slide through the signal. Since both axes are linked, both slide at the same time. The limits of the *y*-axis in both axes are adjusted automatically according to the data within the selected range. However, if the "Fix *y*-limits" checkbox is selected, the range of the *y*-axis of the tachogram is fixed to the current limits. Since some users favor a tachogram that displays the RR-intervals, while others favor HR values, we made it possible to switch between the two.

An ECG recording with multiple channels might result in axes that become unclear. Therefore, R-DECO enables the user to switch view between different channels. This way the user can select one, multiple or all channels. If the channel labels are not present in the signal file, R-DECO names and numbers the channels itself: Channel 1, Channel 2, etc.

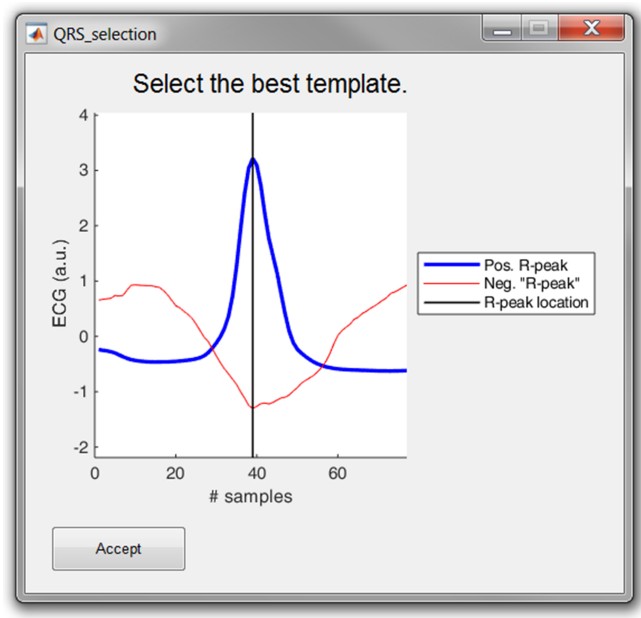

**Figure 8 The template selection window.** The user can select either the positive or "negative" R-peak template and can shift the location of the R-peak if necessary.

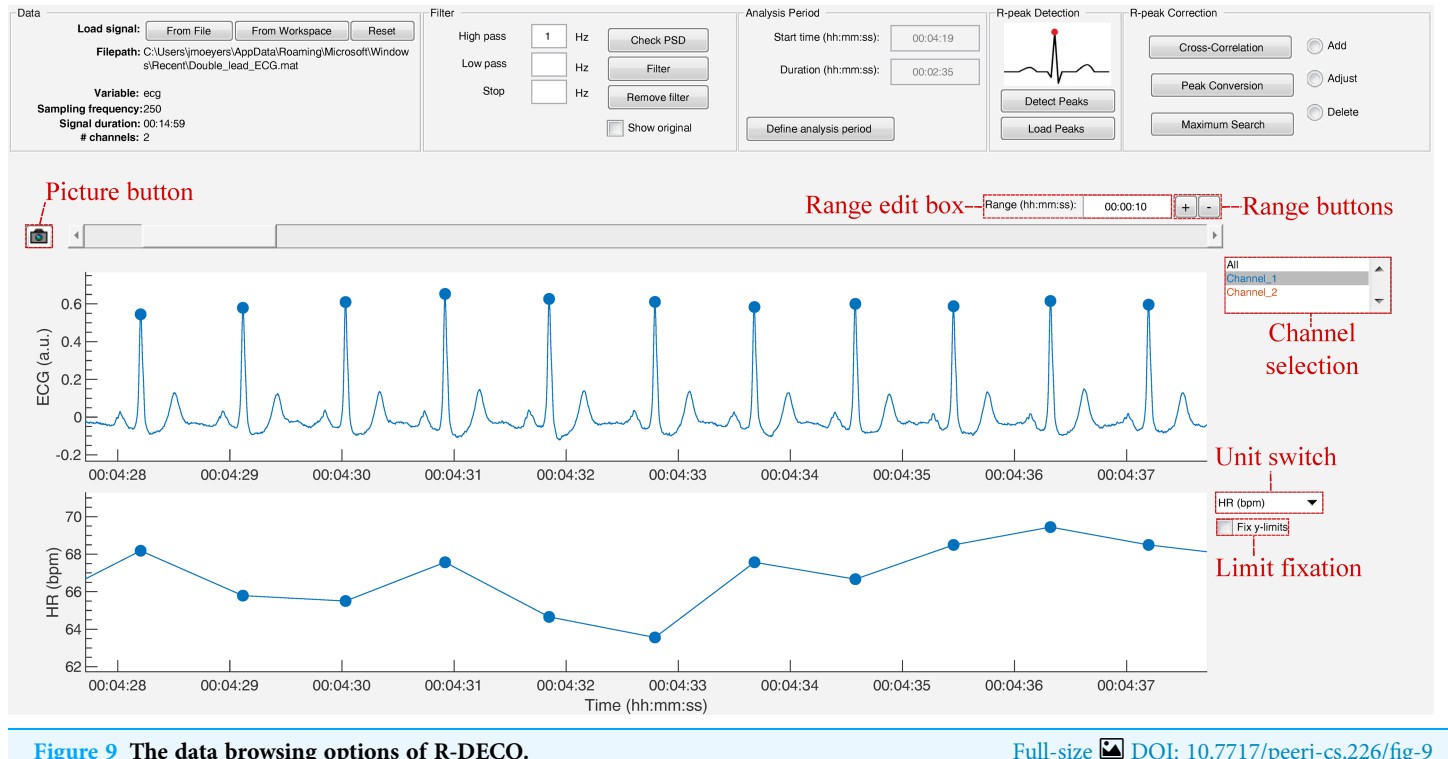

**Figure 9 The data browsing options of R-DECO.**

However, it also provides the possibility to change these names. When the user clicks twice on a channel name in the listbox, a dialog box pops up that allows the user to choose a new name. This will be adjusted in the listbox, and also in the output files.

An extra feature is the possibility to take "pictures" of the axes. This saves both axes in a format of choice, without including any of the buttons or bars from the user interface. In order to design the axes to the user's taste, R-DECO allows to change the line colors and the grid lines.

## Preferences

The analysis settings of R-DECO can be adjusted via the Preferences menu. Note that the changes only apply for the current session and are not saved for the next session. This will be adjusted in a future release.

The Preference menu can be divided in four segments: Power Spectrum, Filter, Detection and Correction

- Power Spectrum: All variables of the Welch method can be adjusted here.
- Filter: The type and order of the filter can be defined here.
- Detection: The default input parameters for the R-peak detection algorithm can be adjusted here. Whenever the default button is pressed in the parameter selection window, see Fig. 7, all parameters are reset to the values defined in this segment.
- Correction: For all correction methods, the user can define the window size in which the new R-peak is supposed to be located.

## SAMPLE RUN

As a sample run, we used a 24 h digital Holter signal that was recorded from a male subject with ischaemic heart disease. The idea was to investigate the temporal evolution in BVR before spontaneous non-sustained ventricular tachycardia (nsVT). Before analysis could be performed, the nsVT episodes needed to be identified and the R-peaks needed to be detected.

Once the signal was loaded and the sampling frequency was defined, we first had a look at the power spectrum. According to the plot, no power line interference was present, since we could not observe a peak at 50 or 60 Hz. However, it was clear that most of the power was situated in the lower frequency bands. This could indicate the presence of baseline wander. Therefore, we high-pass filtered the signal with a cut-off frequency of 0.66 Hz. The result in the power spectrum can be observed in Fig. 10.

Next, the nsVT episodes needed to be identified. However, the time stamps of the episodes were not known. Therefore, it was necessary to detect the R-peaks first. This way the nsVT episodes could be identified from the tachogram. An example of an nsVT episode, taken with the picture button, can be observed in Fig. 11.

Based on the example signal, we selected an envelope size of 300 ms, which provided the best results for this signal. This envelope size ensures the enhancement of the QRS-complexes, without skipping any beats. Additionally, we indicated that no post-processing of the RR-intervals is wanted, since we wanted to be able to detect nsVT segments as well.

The nsVT episodes were identified based on the resulting RR-intervals. From the start of one of these episodes, we selected 30 consecutive heartbeats (*Thomsen et al., 2004*). Only normal-to-normal intervals should be taken into account for BVR-analysis. Hence,

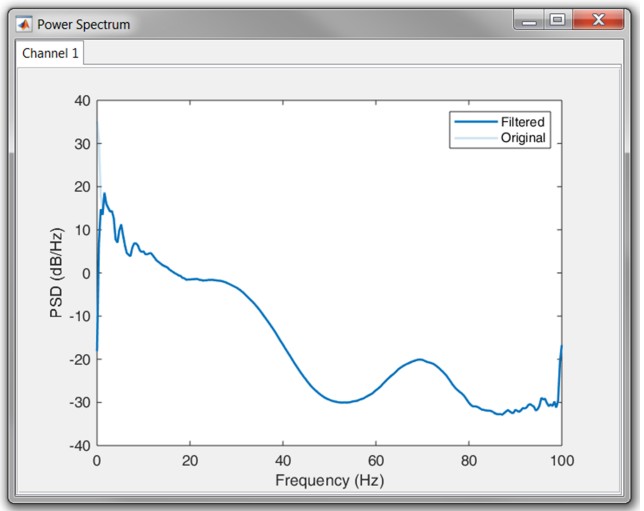

**Figure 10 Result of the high pass filtering in the power spectrum.**

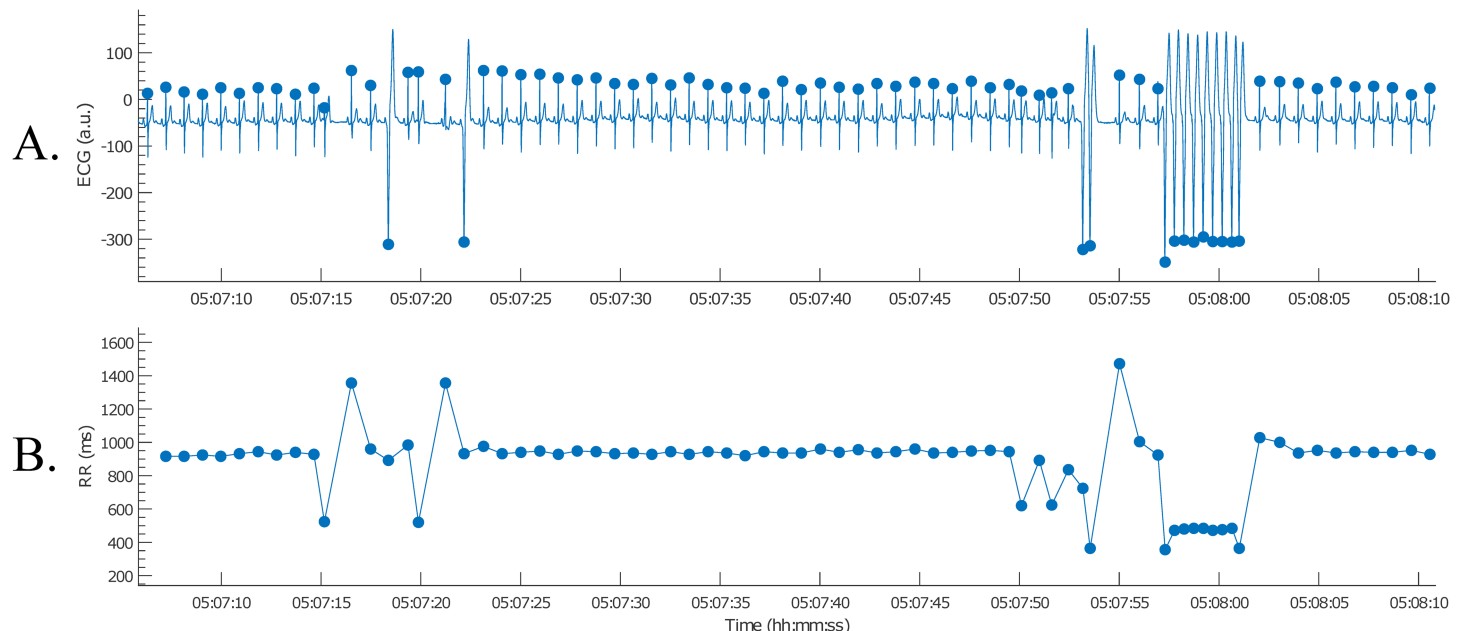

**Figure 11 Example of an nsVT segment with correction (A); the resulting tachogram is shown in (B).**

(ventricular) ectopic and post-extrasystolic beats were removed for further analysis, as can be observed in Fig. 12.

After the RR-intervals in the wanted analysis window were selected, the analysis results were saved. This was done by selecting "Save Results" on the menu bar and entering a file name. The results are then saved as a MATLAB file and can be loaded for the following BVR-analysis. Note that the results can also be exported as an Excel file.

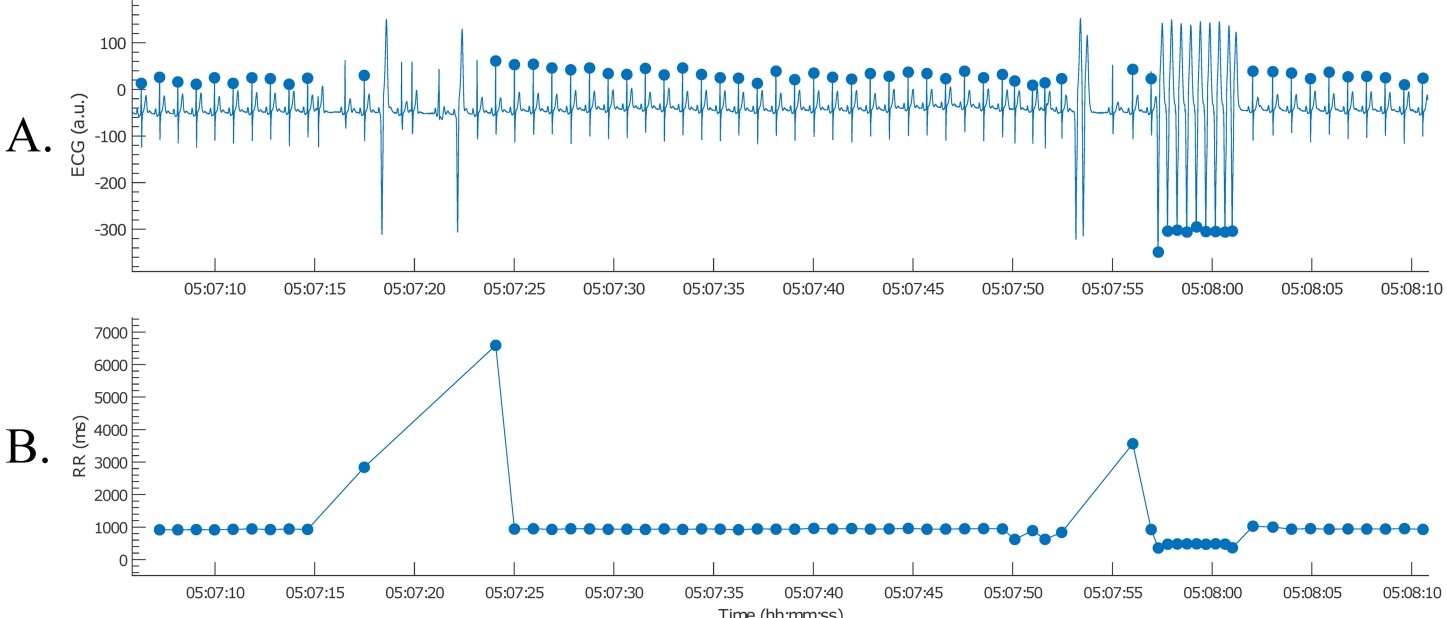

**Figure 12 Example of an nsVT segment with correction (A); the resulting tachogram is shown in (B).**

## POTENTIAL OF FUTURE GROWTH

R-DECO is the first step toward a complete ECG processing tool. At the moment, it focusses on accurate R-peak detection and intuitive correction options. Therefore, it is a complementary tool for existing HRV-analysis toolboxes, which tend to focus on the computation of HRV metrics from RR-intervals. Although some toolboxes already provide the possibility to detect R-peaks, their possibility to correct R-peak annotations is rather limited (*Pichot et al., 2016*; *Rodenhauser et al., 2018*; *Vicente et al., 2013*). Additionally, to the best of our knowledge, none of the existing toolboxes provide filtering, R-peak detection and correction all together.

The main advantage of R-DECO is the easy-to-use, intuitive GUI. All actions are performed in one window, which simplifies the use and reduces the learning time.

Several extra features are being developed and will be released in future versions. We intend to add support for other input file formats, such as hierarchical data format 5 files, general data format files, etc. However, most improvements will be in the number of analysis options. Some of the first extra analysis options will be automatic signal quality detection, EDR and HRV-analysis.

## CONCLUSION

R-DECO is a MATLAB based GUI for detecting and correcting R-peaks in ECG signals. The goal of R-DECO is to provide a complete workflow from the raw signal to the tachogram. It includes an accurate R-peak detection algorithm, the performance of which is comparable to the state-of-the-art, and allows the user to graphically correct wrong or missing detections. Additionally, R-DECO supports a variety of ECG input file formats,

which allows the processing of recordings directly from the recording device. This makes it a tool that can be used both by engineers, and clinicians.

We included some basic pre-processing options, such as three filters and the possibility to select an analysis window. The analysis results can be exported to the MATLAB workspace or Excel for later analysis.

R-DECO is available free of charge and can be downloaded from https://gitlab.esat. kuleuven.be/biomed-public/r-deco.

## ACKNOWLEDGEMENTS

Matthew Amoni is a doctoral fellow of the Research Foundation-Flanders (FWO). Carolina Varon is a postdoctoral fellow of the Research Foundation-Flanders (FWO). Rik Willems is a Senior Clinical Investigator of the Research Foundation-Flanders (FWO).

### Funding

This work was supported by the Bijzonder Onderzoeksfonds KU Leuven (BOF): C24/15/036, C24/18/097; Agentschap Innoveren & Ondernemen (VLAIO): STW 150466 OSA+, O&O HBC 2016 0184 eWatch; Belgian Foreign Affairs-Development Cooperation: VLIR UOS programs (2013-2019); EU: 26077, 766456, 813120, 813483. The funders had no role in study design, data collection and analysis, decision to publish, or preparation of the manuscript.

### Grant Disclosures

The following grant information was disclosed by the authors:
Bijzonder Onderzoeksfonds KU Leuven (BOF): C24/15/036, C24/18/097.
Agentschap Innoveren & Ondernemen (VLAIO): STW 150466 OSA+, O&O HBC 2016 0184 eWatch.
Belgian Foreign Affairs-Development Cooperation: VLIR UOS programs (2013–2019).
EU: 26077, 766456, 813120, 813483.

### Competing Interests

The authors declare that they have no competing interests.

### Author Contributions

- Jonathan Moeyersons conceived and designed the experiments, performed the experiments, analyzed the data, contributed reagents/materials/analysis tools, prepared figures and/or tables, performed the computation work, authored or reviewed drafts of the paper, approved the final draft.
- Matthew Amoni conceived and designed the experiments, authored or reviewed drafts of the paper, approved the final draft, provide inside in the wishes of cardiologists.
- Sabine Van Huffel conceived and designed the experiments, authored or reviewed drafts of the paper, approved the final draft.

- Rik Willems conceived and designed the experiments, authored or reviewed drafts of the paper, approved the final draft, provide inside in the wishes of cardiologists.
- Carolina Varon conceived and designed the experiments, prepared figures and/or tables, performed the computation work, authored or reviewed drafts of the paper, approved the final draft.

### Data Availability

The codes are available at: Moeyersons J, Amoni M, Van Huffel S, Willems R, Varon C, R-DECO: An open-source MATLAB-based graphical user interface for the detection and correction of R-peaks, UNDER REVISION, 2019. https://gitlab.esat.kuleuven.be/biomed-public/r-deco.

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
