# Peer review of "R-DECO: an open-source Matlab based graphical user interface for the detection and correction of R-peaks"

_PeerJ Computer Science, doi:10.7717/peerj-cs.226_

## Round 0.1 · original submission · Major Revisions

The paper is well written and presents a tool that looks potentially very useful. However, there are some problems that preclude publication at this point. Please address my comments and the reviewers' ones.

Elgendi et al. (2014) finds that there are several methods that outperform Pan-Tompkins according to both, sensitivity and positive predictive value. The tool needs to include a method that is at the state-of-the-art. Alternatively, there needs to be a convincing explanation in the paper why these other methods are not suitable for the tool.

Regardless of this issue, the paper does not make it sufficiently clear why your modified version of the Pan-Tompkins algorithm is used. The results quoted (lines 123-127) do not show an advantage. Moreover, the paper states that they were obtained using an envelope width of 300ms. How was this envelope width obtained? By choosing the value that produced the best results on the test data? This would yield optimistically biased performance estimates. Perhaps it is possible to at least show how sensitive performance is to the choice of width?

The software is available free of charge, but only "upon request". This condition makes publication of this paper significantly less attractive.

The URL for the software in the PDF does not work. I managed to guess the correct URL (http://homes.esat.kuleuven.be/~jmoeyers/R_DECO.zip). There is no licensing information in the .zip file. Please consider using an open-source license such as the GPL.

PeerJ CS is mainly targeted at a computer science audience. Please include an illustration of ECG events such as the one in Figure 1 in Elgendi et al. (2014).

"Although it was developed in the eighties, it has proven to outperform many more elaborate algorithms." -- This is misleading. There are actually a number of algorithms in Elgendi et al. (2014) that outperform this old method.

"A slightly modified version of the search-back procedure" -- Please specify the exact modification.

"necessary, because" -- Remove comma.

"In Table 1 we listed" -> "In Table 1 we list"

"PVC" -- Please explain what this is.

Please explicitly state in the text how R-peaks are represented in Figure 2 (small filled circles?).

"in the field of Holter" -- ?

"also a zero phase notch filter is included" -> "a zero phase notch filter is also included"

"however this is not always the case" -> "but this is not always the case"

"As can be seen from Fig 4" -> "In Fig 4"

"appreciate the changes" -> "enact the changes"

"from e.g. an" -> "from, e.g., an"

"on the first sheet a general overview of the file" -> "a general overview of the file on the first sheet"

"The y-axis range in both axes" -- ?

"provides the user to switch view" -> "enables the user to switch view"

Please include (a) screenshot(s) to illustrate what is described in lines 347-350.

"Note that you can also export the results as an Excel file" -> "Note that the results can also be exported as an Excel file"

"amount of analysis options" -> "number of analysis options"

"comparable to literature" -> "comparable to the state-of-the-art" ???

Please check the references, e.g., for incorrect typesetting of acronyms.

Reviewer 1 ·

Basic reporting

Overall the manuscript is well written, while I found the description of Decision (Line 97-103) is not clear, which needs to be further improved. Perhaps a figure would help.

Experimental design

As mentioned above, please provide details in the Decision step on Page3.

Validity of the findings

The authors mentioned that the software is freely available, but the provided link (http://homes.esat.kuleuven.be/jmoeyers/R DECO.zip) seems to be not working.

Reviewer 2 ·

Basic reporting

The paper is well written and easy to follow.

Experimental design

no comment

Validity of the findings

no comment

Additional comments

The paper presents the description of an useful open-source Matlab based graphical user
interface for the detection and correction of R-peaks, the QRS complex detection is based on and envelope-based procedure used in combination with an adapted version of a traditional threshold-based QRS detector. The paper is well written and easy to follow. The proposals are good, and the results are interesting. However, some minor sections of the paper need to be clarified.

Author stated that decrease in performance is generally due to poor signal quality. Therefore it should be interesting to have a discussion about the Noise Tolerance performance of the proposed QRS Detection
algorithm against noise. What is the effect of the SNR on the proposed tool? Can you tell us what is the threshold limit of SNR? under which SNR the tool can still have good results, and in which conditions the result may be seriously contaminated. An example illustrating this point should interesting for the reader.

Can you quantify the noise effects in the QRS detector by using the noise stress test recommended by the ANSI/AAMI EC38 standard using the records from the MIT-BIH Noise Stress Test Database?

---

## Round 0.2 · Minor Revisions

In my view, the submission is basically ready to be published. However, I would like to see the code in a more permanent location *before* I click the accept button. If Leuven does not have an institutional repository for software, you should look at Zenodo or Figshare:

https://zenodo.org/

or

https://mozillascience.github.io/code-research-object/

A couple of other little remaining issues:

"Although it was developed in the eighties, it achieved comparable performance to many 52 more elaborate algorithms.” -> “… achieves … “ ??? [Also, should this be supported by a reference to Elgendi’s comparison?]

Please add a one-sentence explanation of the term "RR interval" when it is first introduced.

Please specify the role of the user-specified “Average heart rate” in your tool (it occurs in Figure 7).

Typos:

"It achieved a 99.60%” -> “It achieves a 99.60%”

qrs, ecg, etc., still occur in lowercase in the bibliography!

---

## Author Rebuttal · Round 0.2

# Summary of Changes

### Artefact detection and quality assessment of ambulatory ECG signals
### Nr: 37482

J. Moeyersons, M. Amoni, S. Van Huffel, R. Willems, C. Varon

We have prepared a new version of our manuscript 37482 taking into account the comments of the reviewers. Additions to the manuscript are now highlighted in blue and the main changes are summarized as follows:

- Added the results of an experiment showing the sensitivity of the performance to the window width of the enveloping procedure.

- Added licensing information in the code files.

- Updated the URL.

- Tested the performance of the algorithm under noisy conditions.

- Improved the quality of the figures.

## Answers to the Reviewers

### Editor

*- Elgendi et al. (2014) finds that there are several methods that outperform Pan-Tompkins according to both, sensitivity and positive predictive value. The tool needs to include a method that is at the state-of-the-art. Alternatively, there needs to be a convincing explanation in the paper why these other methods are not suitable for the tool. Regardless of this issue, the paper does not make it sufficiently clear why your modified version of the Pan-Tompkins algorithm is used. The results quoted (lines 123-127) do not show an advantage.*

Thank you for pointing that out. The mentioned disadvantage, is actually an advantage of the tool.

The R-peak detection method that is implemented was developed in our group (Varon et al. 2015) and it has shown to perform comparably to other state-of-the-art algorithms. However, this does not mean that other methods are not suitable. In the paper we currently mention the following:

"Since some devices have built-in QRS detection algorithms and some researchers have their own preferred QRS detection algorithm, the software allows to load R-peak locations."

Hence, other methods could be used to detect the R-peaks beforehand and the tool could be used to correct possible errors. Moreover, since the tool is open-source, the user also has the possibility to implement his/her preferred method. The detection of R-peaks is only a single line of code, so if the output of the other method is formatted similarly to the implemented method, the user can simply replace it.

- *Moreover, the paper states that they were obtained using an envelope width of 300ms. How was this envelope width obtained? By choosing the value that produced the best results on the test data? This would yield optimistically biased performance estimates. Perhaps it is possible to at least show how sensitive performance is to the choice of width?*

Since the envelope width was empirically selected, it might indeed be true that the performance is optimistically biased. Therefore we have conducted an experiment to evaluate the sensitivity of the performance to the choice of window width.

We have computed the average PPV and Sensitivity for the entire dataset for window widths ranging from 200 ms to 400 ms. This was done both with and without post-processing. The results of this experiment are described below and this section was also added to the paper.

"As mentioned previously, the pre-processing consists of a flattening step of the ECG with a user-defined window width. To evaluate the sensitivity of the performance to the choice of the width we have tested multiple window widths. As can be observed from Fig 1, comparable results were obtained for window widths between 250 and 350 ms."

[Figure]

Figure 1: **Sensitivity of the performance to the choice of window width.** Blue: with post-processing, Red: without post-processing, A window width between 250 and 350 ms results in the best performance.

- *The software is available free of charge, but only "upon request". This condition makes publication of this paper significantly less attractive.*

This was adjusted to "free of charge".

- *The URL for the software in the PDF does not work. I managed to guess the correct URL (`http://homes.esat.kuleuven.be//~jmoeyers/R_DECO.zip`).*

Thank you for noticing. We corrected the URL.

- *There is no licensing information in the .zip file. Please consider using an open-source license such as the GPL.*

A license file (COPYING) was added in the .zip file. Additionally, also a copyright section was added at the beginning of each script.

- *PeerJ CS is mainly targeted at a computer science audience. Please include an illustration of ECG events such as the one in Figure 1 in Elgendi et al. (2014).*

We have a added a general figure (Fig 1) which shows all present waveforms and interesting intervals of a heartbeat as recorded by an ECG.

[Figure]

Figure 2: **A normal heartbeat as recorded by an ECG.** The QRS-complex can be observed in the center. The detection of this complex is crucial for almost all ECG analysis algorithms.

- *"Although it was developed in the eighties, it has proven to outperform many more elaborate algorithms." – This is misleading. There are actually a number of algorithms in Elgendi et al. (2014) that outperform this old method.*

This was adjusted to: "Although it was developed in the eighties, it achieved comparable performance to many more elaborate algorithms."

When looking at the paper of Elgendi et al. (2014) it is indeed true that stating that it outperforms many other methods is an exaggeration. It might be better to state that it performs comparably.

- *"necessary, because" – Remove comma.*

This is adjusted.

- *"In Table 1 we listed" -¿ "In Table 1 we list"*

This is adjusted.

- *"PVC" – Please explain what this is.*

Thank you for noticing. We have added the explanation of PVC's:

"This might be explained by the extremely high percentage of premature ventricular contractions (PVC) present in the recording, almost 15%."

- *Please explicitly state in the text how R-peaks are represented in Figure 2 (small filled circles?).*

The mentioned figure and caption is adapted to:

[Figure]

Figure 3: **The graphical user interface of R-DECO.** The user interface can be divided in five segments: A.) Data, B.) Filter, C.) Analysis Period, D.) R-peak Detection and F.) R-peak Correction. F.) shows an ECG signal with the detected R-peaks (small filled circles) and G.) shows the resulting tachogram.

- *"in the field of Holter" – ?*

This is adjusted to:

"This file format was developed to facilitate data exchange and research in the field of Holter ECG analysis [9]."

- *"also a zero phase notch filter is included" -¿ "a zero phase notch filter is also included"*

This is adjusted.

- *"however this is not always the case" -¿ "but this is not always the case"*

This is adjusted.

- *"As can be seen from Fig 4" -¿ "In Fig 4"*

This is adjusted.

- *"appreciate the changes" -¿ "enact the changes"*

This is adjusted.

- "from e.g. an" -¿ "from, e.g., an"

This is adjusted.

- "on the first sheet a general overview of the file" -¿ "a general overview of the file on the first sheet"

This is adjusted.

- "The y-axis range in both axes" – ?

This is adjusted to:

"The limits of the y-axis in both axes are adjusted automatically according to the data within the selected range."

- "provides the user to switch view" -¿ "enables the user to switch view"

This is adjusted.

- Please include (a) screenshot(s) to illustrate what is described in lines 347-350.

We changed the current figure and text to the following:

"Next, the nsVT episodes needed to be identified. However, the time stamps of the episodes were not known. Therefore, it was necessary to detect the R-peaks first. This way the nsVT episodes could be identified from the tachogram. An example of an nsVT episode, taken with the picture button, can be observed in Fig. 4.

[Figure]

Figure 4: **Example of an nsVT segment without correction.** The resulting tachogram is shown in B.

Based on the example signal, we selected an envelope size of 300 ms, which provided the best results for this signal. This envelope size ensures the enhancement of the QRS-complexes, without skipping any beats. Additionally, we indicated that no post-processing of the RR-intervals is wanted, since we wanted to be able to detect nsVT segments as well.

The nsVT episodes were identified based on the resulting RR-intervals. From the start of one of these episodes, we selected 30 consecutive heartbeats (Thomsen et al. 2004). Only normal-to-

[Figure]

Figure 5: **Example of an nsVT segment with correction.** The resulting tachogram is shown in B.

normal intervals should be taken into account for BVR-analysis. Hence, (ventricular) ectopic and post-extrasystolic beats were removed for further analysis, as can be observed in Fig. 5."

- *"Note that you can also export the results as an Excel file" -¿ "Note that the results can also be exported as an Excel file"*

This is adjusted.

- *"amount of analysis options" -¿ "number of analysis options"*

This is adjusted.

- *"comparable to literature" -¿ "comparable to the state-of-the-art" ???*

This is adjusted to:

    "It includes an accurate R-peak detection algorithm, the performance of which is comparable to the state-of-the-art, and allows the user to graphically correct wrong or missing detections."

- *Please check the references, e.g., for incorrect typesetting of acronyms.*

Thank you for noticing. The references were checked and adjusted where necessary.

### Reviewer: 1

- *I found the description of Decision (Line 97-103) is not clear, which needs to be further improved. Perhaps a figure would help.*

We have added the following section in the paper:
    "A graphical representation of this process is shown in Fig. 6.

- *The authors mentioned that the software is freely available, but the provided link (http://homes.esat.kuleuven.be/̃ DECO.zip) seems to be not working.*

[Figure]

Figure 6: **Procedure to select R-peaks** The resulting flat ECG denoted F is indicated by the black line. The samples with an amplitude lower than the sample 80 ms further are indicated by the magenta circles, with the last sample indicated by the green circle. The search window is indicated by the green line. The selected R-peaks are indicated by the black circles. A.u. stands for arbitrary units.

Thank you for noticing. We have added the correct URL:

> http://homes.esat.kuleuven.be//∼jmoeyers/R_DECO.zip

## Reviewer: 2

*- Author stated that decrease in performance is generally due to poor signal quality. Therefore it should be interesting to have a discussion about the Noise Tolerance performance of the proposed QRS Detection algorithm against noise. What is the effect of the SNR on the proposed tool? Can you tell us what is the threshold limit of SNR? Under which SNR the tool can still have good results, and in which conditions the result may be seriously contaminated. An example illustrating this point should be interesting for the reader.*

In the paper we currently mention the following: "While these results are very promising, we can also observe that for some recordings only moderate detection results are achieved. This decrease in performance is generally due to poor signal quality, unusual morphology or stretches of extremely irregular rhythms."

However, poor signal quality was a bad selection of words. What we meant to say is loss of signal. Therefore, we changed the segment to the following:

"While these results are very promising, we can also observe that for some recordings only moderate detection results are achieved. This decrease in performance is generally due to loss of signal, unusual morphology or stretches of extremely irregular rhythms."

*- Can you quantify the noise effects in the QRS detector by using the noise stress test recommended by the ANSI/AAMI EC38 standard using the records from the MIT-BIH Noise Stress Test Database?*

The noise tolerance of the algorithm was evaluated with the MIT-BIH Noise Stress Test Database (Moody et al. 1984). From Fig. 7 we can observe that both median PPV and sensitivity remained around 100% above a Signal-To-Noise-Ratio (SNR) of 6 dB. Below this threshold the performance of the algorithm decreased significantly.

[Figure]

Figure 7: **Quantification of noise effects** Thick line: Median value, Grey zone: 25th to 75th percentile.

We have added the following segment to the paper:

"The noise tolerance of the algorithm was evaluated with the MIT-BIH Noise Stress Test Database (Moody et al. 1984). We observed that both median PPV and sensitivity remained around 100% above a Signal-To-Noise-Ratio (SNR) of 6 dB. Below this threshold the performance of the algorithm decreased significantly."

---

## Round 0.3 · accepted · Accept

Thank you for making the requested changes. I enjoyed reading through your paper and hope that the PeerJ CS audience will too!